# Distress as a Function of Social Exclusion and Assertiveness among Homosexual/Bisexual People

**DOI:** 10.3390/ijerph21050633

**Published:** 2024-05-16

**Authors:** Chau-kiu Cheung, Eileen Yuk-ha Tsang

**Affiliations:** Department of Social and Behavioural Sciences, College of Liberal Arts and Social Sciences, City University of Hong Kong, Hong Kong, China; eileen@cityu.edu.hk

**Keywords:** distress, social exclusion, assertiveness, conflict theory, LGB

## Abstract

Homosexual (lesbian or gay) and bisexual (i.e., LGB) people tend to suffer from social exclusion and thus distress. To prevent or relieve distress, the people’s assertiveness about justice and rights is an advocated means, but its effectiveness is uncertain, considering possible conflict with social exclusion. To clarify the effectiveness, this study analyzed data collected from 189 Chinese LGB adults in Hong Kong, which is a special administrative region of China generally Westernized and liberal to sexual orientation. Controlling for prior distress reported, the analysis showed that distress was lower when assertiveness was higher or social exclusion experienced was lower. However, distress was higher when both assertiveness and social exclusion experienced were higher. The higher distress implies a conflict between assertiveness and social exclusion to raise distress. It also implies the need to avoid conflict when promoting assertiveness and eliminating social exclusion to prevent distress in LGB people.

## 1. Introduction

Homosexual (lesbian or gay) and bisexual (i.e., LGB) people suffer more distress than heterosexual people, possibly due to the former’s socially excluded position [1,2]. Social exclusion incorporates resentment, rejection, resistance, and discrimination against a person [3]. The exclusion has thereby been responsible for the distress of LGB people [4]. Under such a socially excluded context, it is natural for LGB people to assert their rights [5]. Their assertiveness is manifest in fighting for rights, striving for justice, and confronting conservatives [5,6]. Despite its empowering effect, assertiveness is also distressful due to its generation of conflict with others, according to conflict theory [5]. Such theory maintains that conflict, as a clash of antagonistic forces from others, foments various problems, including distress [7,8]. Accordingly, assertiveness is likely to be distressful, particularly under social exclusion, as assertiveness and social exclusion represent antagonistic forces impinging on each other to provoke a conflict. Given this likelihood, the effect of assertiveness on distress is thus uncertain and needs empirical clarification. Concerning theory, the clarification shows whether assertiveness represents a distressing conflict or coping to resist and mitigate stress on people with alternative sexual orientations [1,5]. For this clarification, the present study analyzes survey data obtained from Chinese LGB people in Hong Kong, which is a special administrative region of China.

The study of Chinese LGB people from Hong Kong, a Westernized metropolis, would generate findings that are compatible with and informative to knowledge development in the Western world as well [9]. Meanwhile, Hong Kong shares some Chinese or Eastern characteristics that enable it to be a bridge for knowledge interchange and generalization for the whole world [10]. That is, Hong Kong is a valuable research site that generates knowledge that is useful to both the West and East. Specifically, knowledge is useful for both Hong Kong and the West because the Westernization of Hong Kong has developed its emphasis on liberalism, which champions people’s diverse sexual orientations and rights [11]. Thus, the sexuality and partnership of LGB people are not illegal there. This echoes the liberal inclination of Hong Kong society [9]. Nevertheless, same-sex marriage is still not legally registerable in Hong Kong, largely following the Chinese or Eastern tradition [11]. This tradition, nonetheless, fuels social exclusion against LGB people, whereas liberalism discourages the exclusion [12]. By contrast, liberalism encourages assertiveness about people’s rights [13]. Overall, the co-occurrence of liberalism and Chinese tradition in Hong Kong sustains both social exclusion and assertiveness there. Hong Kong is thus crucial for examining the joint impacts of social exclusion and assertiveness on distress in LGB people. Notably, LGB people suffer from distress and social exclusion, possibly compounded by the heteronormative tradition of Chinese origin [11,14].

Distress, as a negative emotional state, is noxious, undesirable, and thus in need of prevention and mitigation in LGB people [15]. Notably, distress has fomented suicide and dysfunction [16]. Distress is also detrimental to health [4]. Furthermore, distress heightens demands for support and counseling [17]. Such problems and demands are costly personally and societally, thus prompting practices and related research to prevent distress. Such research has found that while the influences of external or social factors, such as social exclusion on distress, are clear, the influences of personal factors, such as coping or assertiveness, are unclear because of constraint by distress [15,18]. That is, as a distressed person cannot cope well with distress, the preventive or healing effect of coping with distress is questionable. This question thus urges research to control for prior distress when examining personal influences on distress. 

Social exclusion, including rejection and discrimination in social life, is a problem plaguing LGB people and thus in need of tackling and related research [3,18]. According to liberalism, social exclusion is undesirable because it removes personal freedom to live in a society [12]. Notably, social exclusion has impeded one’s deliberation and choice [19]. Social exclusion is also undesirable because of its noxious influences on personal well-being, including anxiety, depression, and distress [4]. More generally, social exclusion has also involved threat, violence, and crime [20]. When imposed on LGB people, social exclusion is notable stress, which is distressing [18]. 

Assertiveness in terms of fighting for rights and justice and against adversaries is noteworthy in LGB people and related research [5,21]. In the face of minority stress, assertiveness is a way of coping [22]. Hence, assertiveness is a natural and effective response to stress, according to the stress model, which maintains social exclusion as stress [23]). However, when assertiveness is forceful, it is also conflictual with others [24]. Assertiveness is thus both empowering and conflictual. Its empowering effect has safeguarded well-being, including self-esteem [21]. Assertiveness is also likely to protect LGB people from the hurt of social exclusion [25]. Hence, assertiveness among LGB people is practically important as a goal for promotion or training [26]. Assertiveness, however, has generated conflicts that intensify radicalism and destruction [27]. Considering these mixed effects, clarifying the effect of assertiveness on distress is vital.

### 1.1. Effects of Social Exclusion and Assertiveness on Distress

Experience of social exclusion is likely to foment distress, whereas assertiveness is likely to dampen distress, according to conflict theory. This theory posits that conflict, competition, or power struggle is disturbing [18,28]. Power or force can stem from different sources, including social class, authority, institution, and collectivity [7]. The influences of power are empowering or disempowering when they enhance or deprive one’s resources respectively. Whereas empowerment is salutary, disempowering is distressful. Disempowering factors have included abuse, aggression or violence, competition, demand, limitation, sanctioning, stigmatization, stress, and threat [14,15,18,29]. A notable example of such power, force, or conflict is class domination, exploitation, oppression, and alienation, which deprive resources and self-actualization [28]. In addition, conflict can occur in myriad interpersonal, intergroup, and even intrapersonal situations and is distressing [30,31]. More specifically, conflict or disempowering force is noxious because of its deprivation, draining, and controlling effects [32]. Hence, conflict emphasized in conflict theory encompasses stress that is discriminating against, oppressive, and stigmatizing people highlighted in the minority stress model [23,33]. Conversely, empowering forces alleviating distress have stemmed from social support, treatment, and training received [34,35]. In addition, personal coping, confrontation, participation in collective action, and other forceful practices are empowering or resisting disempowering to prevent distress or counter the distressing effect [36].

Social exclusion is likely to be distressing to the excluded person because of the disempowering, conflictual force imposed by the exclusion, according to conflict theory. That is, social exclusion represents a conflicting, disempowering, or depriving force [37]. Particularly, social exclusion indicates the stress of discrimination and rejection in the stress model [23]. The conflict thus has fomented distress and eroded life satisfaction in the excluded person [38,39]. Generally, social exclusion has exerted its distressing effect through discrimination, isolation, limitation, prejudice, shaming, stigmatization, and unfair treatment [4,18,40,41]. In addition, shame and loneliness resulting from social exclusion are distressing [42]. Conversely, the distressing effect of social exclusion also stems from its reduction of social acceptance, connection, and support [23,34,43].

Assertiveness is likely to prevent distress due to empowering, considering conflict theory. As such, assertiveness represents a way to gather power or resources [44]. Particularly, coping resources relieve distress, according to the stress model [23]. Power or resources have been preventive of distress [4,40]. Moreover, assertiveness may prevent distress through active coping, confrontation, and control [36]. Justice and rights procured through assertiveness have also impeded distress [18]. In addition, assertiveness has sustained self-confidence and resisted victimization [25]. Distress has diminished with self-confidence and risen with victimization [15,41]. 

However, assertiveness is also likely to breed distress, particularly under social exclusion, concerning conflict theory. Herein, the LGB person’s assertiveness and social exclusion from society are antagonistic forces that generate conflict [8]. As such, conflict results from mutual assertiveness, involving social exclusion and counter-exclusion [45]. Moreover, assertiveness conflicts with social exclusion in opposing stands about justice, rights, and legitimacy [30]. Conflict, competition, or negative social interaction has been a precursor to distress [18,31]. Conversely, supportive social interaction has been preventive of distress [46].

### 1.2. Hypotheses

Direct support for the general and conditional effects of assertiveness on distress in LGB people has hitherto been lacking. Herein, the conditional effect refers to that happening under social exclusion. Based on the review of theory and research in the preceding section [4,18,36,40,41], the following hypotheses about the LGB person are therefore necessary for testing in this study.

Social exclusion experienced is positively predictive of distress.Assertiveness is negatively predictive of distress.Assertiveness is less negatively predictive of distress when social exclusion experienced is higher. That is, the coupling of assertiveness and social exclusion together is positively predictive of distress.

The test of the hypotheses is necessary to reveal effects free of possible confounding due to background characteristics, response characteristics, and prior distress. Most importantly, prior distress is a necessary control factor because of its possible influences on social exclusion experienced and assertiveness, as well as distress later [47]. Without control, notably, the effect of assertiveness is uncertain and probably spurious. Background characteristics, including age, gender, education, and sexual orientation, have commonly made a difference in distress [4,48,49]. These characteristics have also made a difference in assertiveness [6,21,50,51]. Meanwhile, age, gender, and education have also made a difference in the social exclusion experienced [52,53]. Response characteristics, including social desirability and acquiescence or the tendency of high ratings, have affected self-report ratings generally [54].

## 2. Method

With approval by an institutional ethics review committee, a survey of Chinese LGB adults in Hong Kong before the COVID-19 outbreak provided data for analysis. These adults participated in the survey through recruitment by some LGB acquaintances of the researchers. That is, the acquaintances asked their LGB acquaintances to complete the survey.

### 2.1. Participants

The participants were 189 Chinese LGB adults located in Hong Kong through some LGB networks. Key persons in the networks helped invite their network fellows to complete the survey during their gatherings in person (not in a laboratory). Each of them participated in the survey voluntarily, in response to a small incentive, as approved by an institutional ethical review committee. As they were LGB network members, they would not be ashamed of any LGB stigma. They had an average of 33.7 years in age and 15.5 years in education (1 year for each grade increment). Among them, 23.5% were female, 66.5% were male, 78.9% were homosexual, and 21.1% were bisexual. Specifically, 13.2% were lesbian women, 65.9% were gay men, 10.4% were bisexual women, and 10.4% were bisexual men.

### 2.2. Measurement

A self-report questionnaire presented items to measure distress, social exclusion experienced in the last year, and assertiveness in the last year. They measured on a five-level rating scale. The ratings generated scores of 0 for the lowest level, 25 for the second level, 50 for the average level, 75 for the fourth level, and 100 for the highest level. Such a 0–100 scale enabled easy interpretation and comparison without distorting the linear scale [55]. Some of the items employed negative phrasing to minimize the problem of acquiescent response [56]. These items spread within sections for 2017 and 2016 to minimize the influence of the preceding rating on the following rating when they measured the same concept [56]. 

Distress in combined seven items, such as “feeling nervous” and “feeling flurried” during the previous fortnight [57]. It showed a composite reliability coefficient of 0.899, based on confirmatory factor analysis [58].

Prior distress in the last year combined seven items, such as “feeling nervous” and “feeling flurried” in the last year [57]. The retrospective measure was useful, considering the validity of the retrospective measurement of quality of life [59]. Such validity could benefit from the specificity of the annual timeframe [60]. It showed a composite reliability coefficient of 0.870, based on confirmatory factor analysis [58].

Social exclusion experienced in the last year combined four items, such as “society rejecting you” and “society resisting you” in the last year [61]. It showed a composite reliability coefficient of 0.934, based on confirmatory factor analysis [58].

Assertiveness in the last year combined three items, “fighting for rights”, “striving for justice”, and “confronting conservatives” [62]. Its composite reliability was 0.754, based on confirmatory factor analysis [58].

Social desirability in the last year combined three items, such as “being ready to help others” and “being confident in your judgment” in 2016 [63]. It showed a composite reliability coefficient of 0.842, based on confirmatory factor analysis [58].

Acquiescence was the average of all rating items to represent the tendency to rate every item highly. It was a control factor used in statistical analysis [64]. 

### 2.3. Procedure

The survey independently obtained complete data from LGB adults for analysis. The analysis began with confirmatory factor analysis to ensure the factorial validity of distress in the current and last years, social exclusion experienced in the last year, assertiveness in the last year, and social desirability in the last year, in the presence of acquiescence. That is, the analysis fitted a model with five trait factors together with one method factor of acquiescence [64,65]. This model restricted items to load on one of the five trait factors and one method factor. Factorial validity was good when loadings on trait factors were high to show convergent validity and loadings on the method factor were low to show discriminant validity, given the restricted independence of the method factor from the trait factors [64,65]. After this measurement validation, linear regression analysis proceeded to test the hypotheses, with the control for background characteristics, response characteristics, and prior distress. The analysis first tested the main effects specified in Hypotheses 1 and 2. Then, the analysis tested the interaction effect indicated in Hypothesis 3. This test involved interactions computed as pairwise products of assertiveness, social exclusion experienced, and prior distress, using their standard scores to minimize the problem of multicollinearity [66]. A further step of the analysis examined if the hypothesized effects held equally in lesbian, gay, and bisexual statuses. This examination required tests for interaction effects involving the statuses. The examination was worthwhile, considering possible differences in psychology and social experience among the three statuses. For instance, bisexuality draws more social exclusion than homosexuality [67]. Meanwhile, the gay person is more assertive than the lesbian person [68]. These differential conditions may underlie differential effects in the three statuses. In addition, linear regression analysis predicted social exclusion experience and assertiveness with background and response characteristics to show the influences of these characteristics throughout all analyses. 

## 3. Results

Distress, social exclusion experienced, and assertiveness were all substantial in Chinese LGB adults (*M* = 43.1–48.0, see Table 1). Meanwhile, social desirability was moderate (*M* = 57.6).

All the distress, social exclusion experienced, assertiveness, and social desirability demonstrated factorial validity based on confirmatory factor analysis. Accordingly, loadings were high on the five trait factors of distress (*λ* = 0.559–0.696, see Table 2), distress in the last year (*λ* = 0.469–0.698), social exclusion experienced (*λ* = 0.512–0.839), assertiveness (*λ* = 0.429–0.802), and social desirability (*λ* = 0.592–0.676). Loadings on the method factor of acquiescence, which was independent of the trait factors, were generally lower but substantial, suggesting the need for controlling for acquiescence throughout the analysis. Overall, the validation was credible, according to satisfactory goodness-of-fit statistics (*L^2^*(244) = 453, SRMR = 0.053, RMSEA = 0.067, CFI = 0.945) [69]. That is, the model identified the five trait factors in the presence of the independent method factor.

The stage of hypothesis testing, following the validation stage, found support for all three hypotheses. Such support evolved from linear regression analysis controlling for prior distress and background and response characteristics, including social desirability and acquiescence. Nevertheless, only prior distress but not background and response characteristics manifested a significant effect on distress (*β* = 0.808, see Column 1 in Table 3). The analysis was credible due to its high explaining power (*R^2^* > 0.69) and tolerance (>0.285). 

Hypotheses 1 and 2 about the main effects of social exclusion experienced and assertiveness in the last year on distress in the current year obtained support. Accordingly, the effect of social exclusion experienced was significantly positive, and the effect of assertiveness was significantly negative (*β* = 0.139 & −0.153, see Column 1 in Table 3).

Hypothesis 3 is about the interaction effect of social exclusion experienced and assertiveness in the last year on distress in the current year attained support. This interaction effect was significantly positive, even after the control for interactions between prior distress and assertiveness and social exclusion (*β* = 0.084 & 0.095, see Columns 2 and 3 in Table 3). That is, only the effect of the interaction between assertiveness and social exclusion experienced was significant, as the effects of the other two interactions were not significant. Its positive effect indicated that the effect of assertiveness on distress was less negative when social exclusion was higher (see Figure 1). 

A further step in regression analysis showed that the effects of prior distress, assertiveness, social exclusion, and their interactions on distress were not significantly different among lesbian, gay, and bisexual statuses (see Table 4). That is, the effects of prior distress, social exclusion, and the interaction between assertiveness and social exclusion were equally positive on distress in the three statuses. Moreover, the effect of assertiveness on distress was equally negative in the three statuses.

In addition, regression analysis revealed some significant influences of background and response characteristics on social exclusion experienced and assertiveness. Accordingly, the response characteristics of acquiescence and social desirability substantially affected social exclusion and assertiveness reported. Notably, social desirability delivered negative and positive effects on social exclusion and assertiveness, respectively (*β* = −0.229 & 0.212, see Table 5). With the control for the characteristics, age maintained a positive effect on social exclusion (*β* = 0.139). In all, background and response characteristics were requisite control factors in the analyses. 

## 4. Discussion

Analysis showed that in the Chinese LGB adult in Hong Kong, distress was significantly higher when earlier social exclusion experienced was higher and assertiveness was lower, controlling for prior distress. That is, social exclusion increased distress, whereas assertiveness decreased distress. More importantly, the decrease in assertiveness was significantly less when social exclusion was higher. All these effects embody conflict theory, such that social exclusion and assertiveness are antagonistic forces raising and alleviating distress, respectively, and the antagonism generates conflict to escalate distress. In other words, the distress-alleviating force of assertiveness is weaker when the distressing force of social exclusion is stronger. This is because of the conflict generated by the two antagonistic forces. The distressing forces reveal conflict effects emphasized in conflict theory in clarifying stress on people with alternative sexual orientations.

Essentially, assertiveness may not be uniformly helpful in alleviating distress in LGB people. That is, the alleviation depends on social exclusion, such that the alleviation holds when social exclusion is low, and the alleviation vanishes when social exclusion is high. Hence, the alleviation generally holds in Hong Kong (i.e., in terms of the main effect of assertiveness) because social exclusion is not strong. That is, the liberal, tolerant context, prominent in Hong Kong, enables the alleviation of distress through assertiveness. This reflects the case that the liberal context has sustained personal power or effectiveness [70]. In terms of conflict theory, this means that the liberal context does not impose a force to constrain personal power. This context has alleviated distress [71]. Conversely, the socially exclusive context has been debilitating, weakening, or restraining personal power or effectiveness [72]. Such restraint is a manifestation of social force emphasized in conflict theory.

The reduced alleviation of distress by assertiveness under social exclusion also indicates that a direct clash or conflict between the forces of assertiveness and exclusion is distressful. This manifests the distressing effect of conflict [31]. According to conflict theory, the distressing effect results from the intensification of force and struggle by conflict [28,41]. Meanwhile, conflict has been the hotbed for violence or aggression, which is distressful [15]. Conflict has also been confusing and debilitating, thus disabling coping with distress [36,73].

Hence, assertiveness has its strengths and limitations. Its strengths are prominent when social exclusion is low. However, assertiveness is incapable of overcoming social exclusion, as it spurs conflicts under social exclusion. The incapability, according to conflict theory, lies in the weakness of personal assertiveness or even collective action relative to the strength of society and its exclusionary force. Hence, personal assertiveness, at best, makes a difference in the person but not in others or society unconditionally [74]. Assertiveness is thus not just a way to cope with stress personally in the minority stress model, but also one to provoke conflict interpersonally and distress according to conflict theory.

Under social exclusion, non-assertive strategies rather than assertiveness would avoid conflict and its distressing effect. Such strategies include rationalistic ones to strengthen cognition, insight, and behavior to negotiate, compromise, connect, and thus regulate emotions and prevent or relieve distress [34,35], essentially in the face of antagonistic forces, interactional and collaborative, rather than solitary and assertive approaches are effective in preventing conflicts and distress [75]. The former approaches need to maintain peace, respect, and fair exchanges in reciprocal rather than unilateral ways [30]. This follows the approach of conflict theory to preventing and reconciling conflicts by dialectical reasoning and practice, which emphasize understanding the opponent to develop and capitalize on compatibility and complementarity with the opponent [28,41]. Such dialectics need to diminish idealism, competition, and threats, thus preventing conflicts.

The key determinant of the LGB’s distress, nevertheless, is social exclusion, as it also alleviated the effectiveness of the LGB’s assertiveness in preventing or relieving distress. Concerning conflict theory, the strength of social exclusion rests on the force of exclusion from society, which encompasses many people around. Exclusion is forceful because it disconnects relations and deprives access to resources entirely, and relations and resources are precious [37,76]. This force furthermore ruins attachment to and the social identity affiliated with society and replaces them with a noxious label [77]. Social exclusion is, therefore, overwhelmingly debilitating [39]. 

The older LGB person suffered more social exclusion. Considering conflict theory, this is because the older person has less power to satisfy society and resist its exclusion [52]. Such power or resources have included physical and cognitive functioning, up-to-date knowledge, and workability [78]. These resources have prevented social exclusion [79].

## 5. Limitations 

The study has clear limitations in its sampling of LGB adults in a single Chinese metropolis of Hong Kong and self-report survey. Accordingly, the sampling was a non-probability one, unable to represent a nonetheless non-registered population of LGB people. Moreover, only Chinese LGB adults in Hong Kong responded to the survey. They were socially active LGB network members gathered for the survey. The study thereby focuses on a single site in Hong Kong. This seriously limits the generalizability of findings from the study, considering sociocultural variation across the world. In addition, the self-report survey cannot afford optimal validity in measurement and causal inference. The measurement is susceptible to response factors, probably not only social desirability and acquiescence. What is more, the measurement cannot ascertain the timing of distress, social exclusion, and assertiveness. Hence, the internal validity of the study also suffers. To corroborate the findings, future research needs to incorporate better sampling, designs, and measurement. Its goal of sampling is to draw a probability sample to represent LGB people in a place or even the whole world to maximize generalizability. By collecting contextual data across places, future research can examine if contextual factors make a difference in findings obtained from a place, thus gauging their generality. In addition, the survey needs to have a repeated panel design to ascertain predictions based on prior factors. To ensure causation, experimental design manipulating such causal factors as social exclusion and assertiveness can fulfill its complementary role in knowledge creation. To optimize the validity of the measurement, combining measures from multiple sources, including the LGB person and those around the person, can reduce the bias due to subjective measurement. 

To advance theoretical understanding, future research can elaborate the mechanisms of conflict theory to generate the impacts of social exclusion and assertiveness on distress in the LGB person. These mechanisms involve the specification of the forces or power of social exclusion and assertiveness and their antagonism and thus the creation of conflict. The specification of the power needs to tap the number of people involved and its impacts on resources, social relations, social identity, attachment, and labeling. In addition, the conflict or the overwhelming force of social exclusion over the effectiveness of assertiveness and, thus, its power to relieve distress is crucial for scrutiny. 

## 6. Clinical Implications

To prevent distress in LGB people, eradicating social exclusion is key. Concerning conflict theory, the eradication hinges on equalizing or balancing complementary as opposed to antagonistic, conflictual power between LGB people and their possible excluders. This means the enhancement of cooperative and collaborative power, including bonding and bridging social capital, within and between the two groups of people [80]. Such enhancement implies the establishment of superordinate power to reduce social exclusion, antagonism, and conflicts. In addition to the balancing, superordinate power needs to eliminate excuses for exclusion, antagonism, and conflict. This includes mutual problematizing or faulting between the two groups [81]. More fundamentally, superordinate power can prioritize the common identity over the group identity of the two groups [80].

Enhancing the assertiveness of LGB people conditionally is also helpful to prevent their distress and aligns with conflict theory as well. Here, assertiveness means confronting and resisting antagonism rather than avoidance and segregation. The crux is the conditional enhancement under the condition of low social exclusion. This means the avoidance of direct antagonism and conflict between the assertiveness of LGB people and social exclusion. Such conditional enhancement is possible because assertiveness is trainable [34]. Considering conflict theory, the training is to empower LGB people to assert their talents and collaborate with other people to reap common gains. The training would be particularly helpful to older LGB people, who experience greater social exclusion. Without the training, older LGB people are less powerful to avoid social exclusion.

## Figures and Tables

**Figure 1 ijerph-21-00633-f001:**
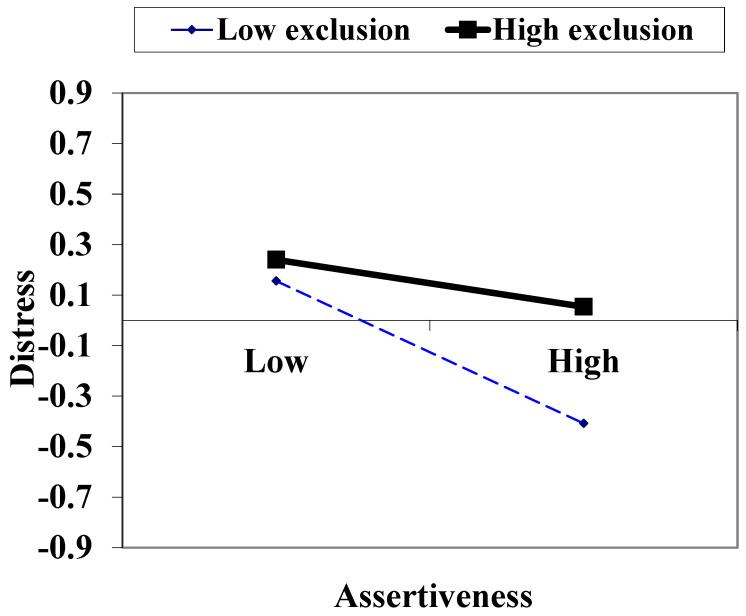
Standard score of distress by high (1 *SD* above *M*) and low levels (1 *SD* below *M*) of assertiveness and social exclusion experienced.

**Table 1 ijerph-21-00633-t001:** Means and standard deviation (*N* = 189).

Variable	Scoring	*M*	*SD*
Age	years	33.7	12.1
Education	years	15.5	3.1
Female	0, 100 (%)	23.5	41.8
Homosexual	0, 100 (%)	78.9	40.5
Social desirability, last year	0–100	57.6	21.1
Acquiescence	0–100	44.3	11.5
Distress, current year	0–100	43.3	20.0
Distress, last year	0–100	43.1	18.6
Assertiveness, last year	0–100	45.4	21.4
Social exclusion experienced, last year	0–100	48.0	23.7

**Table 2 ijerph-21-00633-t002:** Standardized factor loadings on five trait factors and one method factor.

Factor/Indicator	Trait	Method
Distress, current year		
Feeling nervous	0.593	0.544
Feeling flurried	0.696	0.459
Feeling worried	0.668	0.476
Feeling troubled	0.671	0.523
(not) Calming down	0.582	−0.272
(not) Having self-control	0.616	−0.209
(not) Having emotional stability	0.559	−0.213
Distress, last year		
Feeling nervous	0.469	0.562
Feeling flurried	0.620	0.501
Feeling worried	0.605	0.495
Feeling troubled	0.602	0.508
(not) Calming down	0.698	−0.329
(not) Having self-control	0.639	−0.323
(not) Having emotional stability	0.675	−0.366
Social exclusion experienced, 2016		
Society rejecting you	0.819	0.460
Society resisting you	0.671	0.375
Society discriminating against you	0.512	0.422
Society resenting you	0.839	0.434
Assertiveness, last year		
Fighting for rights	0.802	0.503
Striving for justice	0.429	0.519
Confronting conservatives	0.530	0.503
Social desirability, last year		
Being ready to help others	0.592	0.368
Being happy to admit mistakes	0.637	0.447
Treating people with disagreeable opinions courtesy	0.657	0.369
Being confident in your judgment	0.676	0.293

**Table 3 ijerph-21-00633-t003:** Standardized regression coefficients on distress, current year.

Predictor	(1)	(2)	(3)
Age	−0.052	−0.041	−0.049
about the Female	−0.010	−0.006	−0.009
Education	−0.002	0.001	−0.007
Homosexual vs. bisexual	0.020	0.009	0.010
Social desirability, last year	0.079	0.097	0.118
Acquiescence	0.080	0.128	0.093
Distress, last year	0.808 ***	0.801 ***	0.797 ***
Assertiveness, last year	−0.153 **	−0.198 **	−0.205 **
Social exclusion experienced, last year	0.139 **	0.126 *	0.146 **
Assertiveness × Social exclusion experienced, last year		0.084 *	0.095 *
Distress × Assertiveness, last year			−0.061
Distress × Social exclusion experienced, last year			0.055
*R^2^*	0.694	0.699	0.704
*Minimum tolerance*	0.358	0.309	0.285

(1) Main effects only. (2) One interaction effect added. (3) Two more interaction effects were added. * *p* < 0.05. ** *p* < 0.01. *** *p* < 0.001.

**Table 4 ijerph-21-00633-t004:** Additional standardized regression coefficients of interactions with LGB status on distress, current year.

Predictor	Lesbian	Gay	Bisexual
Distress, last year	0.045	0.024	−0.063
Assertiveness, last year	−0.018	0.043	−0.024
Social exclusion experienced, last year	−0.052	0.064	−0.033
Assertiveness × Social exclusion experienced, last year	−0.023	0.023	−0.006
Distress × Assertiveness, last year	−0.006	0.047	−0.051
Distress × Social exclusion experienced, last year	−0.021	0.050	−0.045

No effect of any alternately added predictor was significant at *p* < 0.05.

**Table 5 ijerph-21-00633-t005:** Standardized regression coefficients on social exclusion and assertiveness.

Predictor	Exclusion	Assertive
Age	0.139 *	−0.013
Female	−0.011	0.099
Education	−0.031	−0.016
Homosexual vs. bisexual	0.036	−0.039
Social desirability, last year	−0.229 **	0.212 **
Acquiescence	0.607 ***	0.566 ***
*R^2^*	0.299	0.481
*Minimum tolerance*	0.772	0.772

* *p* < 0.05. ** *p* < 0.01. *** *p* < 0.001.

## Data Availability

Data for the study will be available upon reasonable request.

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
