# Peer review of "Distress as a Function of Social Exclusion and Assertiveness among Homosexual/Bisexual People"

_ijerph, 2024, doi:10.3390/ijerph21050633_

Round 1
Reviewer 1 Report
Comments and Suggestions for Authors
Congratulations, The paper is very interesting! Please can you read the review report for some adjustments?

Author Response
Thanks for your instructive advice. The following outlines major changes highlighted in the revised text.
- Regarding findings in the abstract, the revision rewrites to clarify the findings.
- Regarding references for the review, the revision adds some relevant references cited in the preceding section.
- Regarding participants, the revision clarifies their recruitment as members of some LGB networks in Hong Kong.
Reviewer 2 Report
Comments and Suggestions for Authors
Thank you for the opportunity to read your fascinating work. Coming from the U.S., I am happy to see quality work done with LGB populations. So-called 'cross-cultural' work needs to be widely disseminated.
I had a few questions about wording/clarity:
1) "Furthermore, the effect of assertiveness was significantly less negative when social exclusion was higher." I see in Table 3 that Assertiveness was a negative predictor of current distress and I think that is what is being referred to in the abstract. Perhaps "In regression analyses, personal assertiveness was a negative predictor for current distress such that as assertiveness increased current distress went down." This is only one suggestion that is a bit more straightforward than the current wording.
2: Did the participants come into the lab or did they complete the survey online and thus with the benefit of privacy and nondisclosure? It is not clear from the method section and I think important considering the stigma that is still associated with being LGB.
3) "A self-report questionnaire presented items to measure distress, social exclusion experienced in the last year, and assertiveness in the last year. Based on a five-level rating scale, these items generated scores of 0 for the lowest level, 25 for the second level, 50 for the average level, 75 for the fourth level, and 100 for the highest level. Such a 0-100 scale enabled easy interpretation and comparison, without distorting the linear scale [55]." Were participants asked to rate these items on a scale of 1-100 which was then converted to 0, 25, 50, 75, 100? OR were they asked to rate the items on 1-5 scale which was converted to 0-100? It is not clear from the current method section.
4) Accordingly, the 246 effect of social experience experienced was significantly positive and the effect of asser- 247 tiveness was significantly negative (β = .139 & -.153, see Column 1 in Table 3). 248 Double word here. Line 248
5) "In terms of conflict theory, this means that the liberal context does not impose a force to 301 constrain personal power. Conversely, the socially exclusive context has been debilitating, 302 weakening, or restraining personal power or effectiveness [71]. Such restraint is a mani- 303 festation of social force emphasized in conflict theory." I would love to see more discussion on this point. What is currently written is a bit confusing to me (e.g., 'liberal context does not impose a force') and I am not sure if being in a liberal setting is good? Or not? The personal power that comes from assertiveness is really one of the few things society can't take away- you can have negative consequences for it, of course, but personal power is a big deal. I do wonder, how many of your participants were in relationships? I.e., how many of them were "Out" to their families and communities, especially to those who do not live in HK. I live and work in the rural U.S. and know plenty of people who are openly LGBTQIA+ in safe circles but do not live that way all of the time. Sometimes living your life is the best form of protest? Would that be considered assertive according to your operational definition here?
Overall, I am happy to see a revision of this paper with some clarification of the few points above.
Comments on the Quality of English Language
I gave some brief comments about English language in my main comments. I would like to commend the authors that their use of correct technical and statistical terms was excellent.
Author Response
Thanks for your instructive advice. The following outline major changes highlighted in the revised text.
- Regarding findings in the abstract, the revision rewrites them for clarification.
- Regarding the survey, the revision indicates that participants were LGB network members gathered to complete the survey in person. As the members, they were unlikely to experience stigma for the survey.
- Regarding the scoring, the revision clarifies the scoring of the 1-5 ratings.
- Regarding the liberal context, the revision finds support for lower distress in a more liberal context.
- Regarding the participants' relationships, the revision admits that the participants were active LGB network members in Hong Kong.
- Regarding assertiveness, the revision clarifies that it means confrontation and resistance to antagonism.